# Nanoarchitectonics of Fe-Doped Ni_3_S_2_ Arrays on Ni Foam from MOF Precursors for Promoted Oxygen Evolution Reaction Activity

**DOI:** 10.3390/nano14171445

**Published:** 2024-09-04

**Authors:** Jingchao Zhang, Yingping Bu, Zhuoyan Li, Ting Yang, Naihui Zhao, Guanghui Wu, Fujing Zhao, Renchun Zhang, Daojun Zhang

**Affiliations:** 1Henan Key Laboratory of New Optoelectronic Functional Materials, College of Chemistry and Chemical Engineering, Anyang Normal University, Anyang 455000, Chinarczhang@aynu.edu.cn (R.Z.); 2College of Chemistry, Zhengzhou University, Zhengzhou 450001, China

**Keywords:** Ni_3_S_2_, oxygen evolution reaction, MOF precursors, electrocatalysts

## Abstract

Oxygen evolution reaction (OER) is a critical half-reaction in electrochemical overall water splitting and metal–air battery fields; however, the exploitation of the high activity of non-noble metal electrocatalysts to promote the intrinsic slow kinetics of OER is a vital and urgent research topic. Herein, Fe-doped Ni_3_S_2_ arrays were derived from MOF precursors and directly grown on nickel foam via the traditional solvothermal way. The arrays integrated into nickel foam can be used as self-supported electrodes directly without any adhesive. Due to the synergistic effect of Fe and Ni elements in the Ni_3_S_2_ structure, the optimized Fe_2.3%_-Ni_3_S_2_/NF electrode delivers excellent OER activity in an alkaline medium. The optimized electrode only requires a small overpotential of 233 mV to reach the current density of 10 mA cm^−2^, and the catalytic activity of the electrode can surpass several related electrodes reported in the literature. In addition, the long-term stability of the Fe_2.3%_-Ni_3_S_2_/NF electrode showed no significant attenuation after 12 h of testing at a current density of 50 mA cm^−2^. The introduction of Fe ions could modulate the electrical conductivity and morphology of the Ni_3_S_2_ structure and thus provide a high electrochemically active area, fast reaction sites, and charge transfer rate for OER activity.

## 1. Introduction

In the world today, conventional fossils are increasingly exhausted, and more and more researchers are devoted to developing alternative clean energy to meet increasing human needs [1,2,3,4,5,6,7,8]. Hydrogen energy is considered the most promising clean energy of the future; in all hydrogen preparation technologies, electrochemical water splitting is deemed the most prominent way to acquire green hydrogen [9,10]. Green H_2_ can be prepared by using the electric energy generated by solar energy and wind energy, which are difficult to connect to the grid. Oxygen evolution reaction (OER) is an anodic half-reaction of water splitting, and OER also plays an important role in oxygen-related energy conversion technologies, such as metal–air batteries and fuel cell fields [11,12,13]. In general, active electrocatalysts are needed to promote the OER process to reduce the high overpotential for the electrocatalytic splitting of water [14,15,16,17,18,19,20]. At present, noble metal compounds such as RuO_2_ and IrO_2_ are still effective OER electrocatalysts; however, the high cost and scarce reserves of these state-of-the-art catalysts prevent their widespread application in the future.

Ni_3_S_2_ structure is a notable nickel-based sulfide with a hexagonal structure constructed of Ni-Ni metal bonds and has attracted extensive attention in electrocatalytic and energy storage fields ascribed to its high conductivity, good intrinsic electrocatalytic activity, and low cost [21,22]. Although Ni_3_S_2_ structure is a promising HER and OER electrocatalyst, which exhibits a similar electrocatalytic OER performance to RuO_2_, the performance still does not meet the requirements of large-scale commercial applications, and the overpotential of pure Ni_3_S_2_ is always high because of its sluggish OER kinetics. Researchers are also devoted to optimizing the synergistic electronic modulation to improve the efficiency of OER. Secondary metal doping is an effective strategy to modulate the electronic Ni_3_S_2_ structure. To date, transition metals, including Cr [23], Mn [24,25,26], Cu [27], Co [28], Fe [29,30,31,32], Mo [33], V [34], Sn [35], and Ce [36], have been doped into Ni_3_S_2_ structure to enhance the intrinsic activity; especially, the Fe-Ni bimetallic synergistic effect contributed to forming key active intermediates (FeOOH-NiOOH). Through Fe ions, doping can not only adjust the surface electron distribution but also increase the charge transfer kinetics [37,38].

Ni_3_S_2_ electrocatalysts with different morphologies usually have been synthesized using the solvothermal way, chemical vapor deposition (CVD), and atomic layer deposition (ALD) technology [22,39]. Metal–organic frameworks (MOFs) assembled by metal nodes and organic ligands can be facilely synthetized, and the morphologies and structures can be easily modulated [40,41]. Therefore, micro-/nano-structured transition metal sulfides (TMSs) can be obtained from MOF precursors via a simple sulfurization conversion strategy; especially, MOFs-derived TMSs may inherit the porous structures and exhibit large surface areas [42,43,44,45].

In this work, Fe-doped-Ni_3_S_2_ nanosheet arrays derived from Ni-based MOF precursors as electrode materials were developed. In alkaline solution, the optimized Fe_2.3%_-Ni_3_S_2_/NF electrode exhibits small overpotentials of 233 and 378 mV for OER at 10 and 100 mAcm^−2^, respectively. Moreover, Fe_2.3%_-Ni_3_S_2_/NF delivers good stability for 12 h at 50 mA cm^−2^, and the as-prepared electrode also exhibits no obvious current density decay after 1000 cycles in an alkaline solution.

## 2. Experimental

### 2.1. Reagents and Materials

Nickel (II) 2,4-pentanedionate (Ni(acac)_2_, 95%) was purchased from Alfa Aesar (China) Chemical Co., Ltd. (Shanghai China) Ammonium iron (II) sulfate hexahydrate ((NH_4_)_2_SO_4_·FeSO_4_· 6H_2_O, ≥99.5%), N,N-dimethylformamide (≥99.5%, DMF), ethylene glycol (≥99.5%), and nickel foam (NF) were all purchased from Sinopharm Chemical Reagent Co., Ltd. (Shanghai China) 1,2,4,5-benzenetetra-carboxylic acid (H_4_BTEC) (C_10_H_6_O_8_, >98.0%) and thioacetamide (99%, TAA) were purchased from Shanghai Aladdin Biochemical Technology Co., Ltd. (Shanghai China) Ethanol (≥99.7%) was purchased from Tianjin Fuyu Fine Chemical Co., Ltd. (Tianjin China). It is important to note that all chemicals were of analytical grade and used directly in this work.

### 2.2. Synthesis of Electrocatalysts

Nickel foam (1 × 3 cm^2^) was soaked in 3 M HCl and acetone for 30 min and then washed by sonication to remove impurities on the surface. The thickness of the nickel foam (NF) is 1.6 mm, and the average pore size is 0.2 mm (Appendix A). The catalyst electrodes were prepared via two-step solvothermal processes. The first step is for the synthesis of MOF precursor containing Ni and Fe metal ions and H_4_BTEC on NF substrate. The obtained NF covered with MOF was washed with deionized H_2_O and ethanol and denoted as the FeNi-MOF/NF precursor. The second step is the sulfurization process; considering the low content of Fe ions in FeNi-MOF/NF precursor synthesized in the first step, only Fe_0.6%_-Ni_3_S_2_/NF can be obtained with the absence of Fe ions in the second solvothermal step. However, for the synthesis of the desired electrode with higher Fe ions content from 0.8% to 3.7%, Fe ions must be added extra in the second sulfurization step.

#### 2.2.1. Synthesis of the FeNi-MOF Precursor 

At first, 0.2 mmol Ni(acac)_2_, 0.4 mmol (NH_4_)_2_SO_4_·FeSO_4_·6H_2_O, and 0.8 mmol H_4_BTEC were dispersed in 7 mL DMF, 7 mL ethanol, and 1 mL H_2_O solution by stirring thoroughly for 30 min. Then, the resulting solution was transferred to a 20 mL vial along with a slice of treated NF, followed by processing at 140 °C for 12 h. After cooling, the obtained sample was repeatedly cleaned with deionized water and ethanol and vacuum-dried at 60 °C overnight. The obtained sample was denoted as FeNi-MOF/NF precursor.

#### 2.2.2. Synthesis of the Fe_x_-Ni_3_S_2_/NF

The samples denoted as Fe_0.6%_-Ni_3_S_2_/NF, Fe_0.8%_-Ni_3_S_2_/NF, Fe_2.3%_-Ni_3_S_2_/NF, and Fe_3.7%_-Ni_3_S_2_/NF were prepared by separately dispersive mixing 0.2 mmol TAA with 0, 0.03, 0.06, and 0.12 mmol (NH_4_)_2_SO_4_·FeSO_4_·6H_2_O in 4 mL ethanol, 2 mL ethylene glycol, and 1 mL H_2_O solution under stirring thoroughly for 30 min. Then, the solution and a piece of FeNi-MOF/NF precursor was sealed and heated at 140 °C for 4 h (Figure 1). After cooling to room temperature, the resulting sample was cleaned three times with deionized water and ethanol and vacuum-dried overnight at 60 °C, and the average mass loading of catalyst was ~1.2 mg cm^−2^.

### 2.3. Materials Characterization

The phases of the as-synthesized catalysts were studied by X-ray diffraction (XRD, PANalytical X’ Pert operated at 40 kV and 40 mA) and X-ray photoelectron spectroscopy (XPS, Thermo Scientific K-Alpha, Waltham, MA, USA). Morphologies and elemental distributions of the samples were characterized by scanning electron microscope (SEM, Hitachi SU8010, Tokyo, Japan), transmission electron microscope (TEM, Tecnai G^2^S-Twin F20, Hillsboro, OR, USA), and SEM equipped with energy dispersive X-ray spectroscopy (EDS) analyzer.

### 2.4. Electrochemical Measurements

Electrochemical tests were conducted on an electrochemical workstation (CHI 760E) using a standard three-electrode system in an alkaline solution (1 M KOH). Fe-doped Ni_3_S_2_/NF electrodes were used as working electrodes directly, avoiding the ink preparation and drop-coating process; Hg/HgO and platinum mesh were used as the reference electrode and counter electrode, respectively. The LSV curve was obtained at a sweep speed of 5 mV·s^−1^ in the potential range of 0–0.8 V vs. Hg/HgO without IR compensation, the size of the working electrode was tailored into 1 × 1 cm^2^, and the geometric surface area was used to calculate current density in LSV curves. The Nyquist plots were measured at open circuit potential in the frequency range of 0.1 Hz–1 MHz. The stability test was carried out at a potential of 0.637 V vs. Hg/HgO for 12 h. Multistep chrono-potential and long-term stability measurements were also not compensated for by IR in this work.

## 3. Results and Discussion

The crystalline structures of the synthesized Ni_3_S_2_ were investigated using the powder X-ray diffraction (PXRD) technique. PXRD patterns for the synthesized Fe-doped Ni_3_S_2_ electrodes are exhibited in Figure 1. The series of Ni_3_S_2_ electrodes with various Fe doping amounts have similar diffraction peaks. The two sharp diffraction peaks located at 2θ of 44.4° and 51.9° in all samples belong to the Ni foam. Other diffraction peaks in the PXRD patterns can match with Ni_3_S_2_ (JCPDS No. 044-1418) phase well, and the diffraction peaks located at 21.7, 31.1, 37.8, 49.8, 50.2, and 55.3° were attributable to (101), (110), (003), (113), (211), and (122) crystal planes of rhombohedral Ni_3_S_2_. The results suggest that the pure catalyst was successfully prepared. It is worth mentioning that the different amounts of Fe ions doped did not affect the peak intensity of the Ni_3_S_2_ phase.

The morphology of MOF precursors growing on NF was detected by scanning electron microscopy (SEM) and is shown in Figure 2. Figure 2 presents the SEM images of the Fe-Ni-BTEC MOF precursors for (a) Ni_3_S_2_/NF, (b) Fe_0.6%_-Ni_3_S_2_/NF, (c) Fe_0.8%_-Ni_3_S_2_/NF, (d) Fe_3.7%_-Ni_3_S_2_/NF, and (e,f) different magnifications of Fe_2.3%_-Ni_3_S_2_/NF images. As seen in Figure 2, NF was covered with vertically distributed and irregular MOF nanosheets.

After the second solvothermal sulfurization treatment process (Figure 3), Fe_x_-Ni_3_S_2_/NF series electrodes were obtained. However, if only bare Ni foam was vulcanized, the irregular particles could be obtained (Figure 3a). Using the Fe-Ni-MOF-coated Ni foam as a precursor via sulfurization reaction, some branch structures appeared accompanied by a little thinner and rough Ni_3_S_2_ nanosheet structure (Figure 3b). The yield of branch structure increases with the additional Fe ions in the sulfurization step (Figure 3c–f). The unique branch structure and rough surface are not only conducive to the exposure of the catalytic active sites but also can further accelerate the electrolyte penetration and OER reaction kinetics. Xu et al. reported that Fe ions substituted Ni ions in the (101) plane of Ni_3_S_2_ and decreased the surface energy of the (101) plane [46]. Thus, the introduction of Fe ions in the Ni_3_S_2_ lattice will promote the formation of branch morphology, which was supported by the HRTEM results with the exposure of (101) crystallographic planes. SEM-EDX element mapping also confirms the incorporation of Fe ions in Fe_2.3%_-Ni_3_S_2_/NF. The results of Figure 3g–j show that the as-synthesized Fe_2.3%_-Ni_3_S_2_/NF electrode is composed of Ni, Fe, and S elements, and these elements are uniformly distributed.

In order to further determine the content of Fe element in the synthesized samples, EDX testing was performed on the Fe_x_-Ni_3_S_2_/NF series samples, and the results are shown in Figure 4. Figure 4a shows the EDX measurement results of Fe_0.6%_-Ni_3_S_2_/NF. During the synthesis of Fe_0.6%_-Ni_3_S_2_/NF, only Fe ions were added during the first solvothermal step, and the second step was normal sulfurization treatment. Although Fe element was added during the first solvothermal process, the test results showed that the amount of Fe ions doped was very small, only 0.6%, compared to the composition of other elements. Therefore, in order to increase the amount of Fe ions doped, a small amount of Fe element was added in the second sulfurization process, with the added content being 5%, 10%, and 20% of the Fe content in the first step, respectively. Figure 4b shows the EDX test with the addition of 5% Fe, and it can be seen that the proportion of Fe element did not significantly increase, only the doped 0.8% Fe element. When 10% and 20% Fe ions were added again in Figure 4c,d, the proportion showed a significant increase, with Fe contents of 2.3% and 3.7%, respectively.

TEM characterization of the Fe_2.3%_-Ni_3_S_2_ sample obtained from NF via ultrasonic stripping was conducted and is shown in Figure 5. The low-magnification TEM image in Figure 5a clearly demonstrates the dendritic structure, the HRTEM image exhibits obvious lattice fringes, and the interplanar spacing of 0.40 nm may be attributed to plane (101) of the pristine Ni_3_S_2_ phase.

OER performances of bare Ni foam and the series electrodes, including Ni_3_S_2_/NF, Fe_0.6%_-Ni_3_S_2_/NF, Fe_0.8%_-Ni_3_S_2_/NF, Fe_2.3%_-Ni_3_S_2_/NF, and Fe_3.7%_-Ni_3_S_2_/NF, with different Fe contents were evaluated in O_2_-saturated alkaline medium (1.0 M KOH) and are presented in Figure 6. Figure 6a exhibits the LSV curves for OER of series electrodes and the control sample of bare Ni foam. The redox peak shifting to the higher potentials indicated the Fe incorporation in Ni_3_S_2_ [47]. Compared with Fe-doped Fe_x_-Ni_3_S_2_/NF series electrodes, Ni foam shows the weakest OER performance, Ni_3_S_2_/NF shows a small decrease in overpotential at the current density of 10 mA cm^−2^, and the optimized Fe_2.3%_-Ni_3_S_2_/NF shows the highest activity among the as-prepared electrodes. At a current density of 10 mA cm^−2^, the Fe_2.3%_-Ni_3_S_2_/NF delivers a small overpotential of 233 mV. The value of overpotential is less than that of NF (*η*_10_ = 350 mV), Ni_3_S_2_/NF (*η*_10_ = 340 mV), Fe_0.6%_-Ni_3_S_2_/NF (*η*_10_ = 253 mV), Fe_0.8%_-Ni_3_S_2_/NF (*η*_10_ = 267 mV), and Fe_3.7%_-Ni_3_S_2_/NF (*η*_10_ = 251 mV), and also can compare to the recently reported electrodes such as F-Ni_3_S_2_ (*η*_10_ = 239 mV) [48], Fe-Co_9_S_8_ NM/NF (*η*_10_ = 270 mV) [49], MOF-V-Ni_3_S_2_/NF (*η*_10_ = 268 mV) [50], Mo-doped Ni_3_S_2_ (*η*_10_ = 260 mV) [33], Ni_3_S_2_/NF-4 (*η*_10_ = 242 mV) [51], and Fe_7.2%_-Ni_3_S_2_/NF (*η*_10_ = 295 mV) [52] catalysts (Table 1). Compared with the pure Ni_3_S_2_/NF in Figure 6a, the Fe element-doped Ni_3_S_2_ electrodes have greatly enhanced the activity of OER performances. Further detailed comparison of the overpotentials of series electrodes at 10 and 100 mA cm^−2^ is revealed in Figure 6b, respectively. Both of the results suggest that incorporating Fe ions into the Ni_3_S_2_ host lattice can greatly improve its intrinsic OER catalytic activity. In addition, effective regulation of the doping content of Fe ions has a significant impact on the OER catalytic performance of the as-synthesized Fe_x_-Ni_3_S_2_/NF series electrodes.

Figure 6c demonstrates the Tafel slopes of bare Ni foam, Ni_3_S_2_/NF, Fe_0.6%_-Ni_3_S_2_/NF, Fe_0.8%_-Ni_3_S_2_/NF, Fe_2.3%_-Ni_3_S_2_/NF, and Fe_3.7%_-Ni_3_S_2_/NF electrodes, which were derived from LSV curves with values of 80, 95, 79, 78, 66, and 74 mV dec^−1^, respectively. It can be clearly seen from Figure 6c that the Tafel slope of Fe_2.3%_-Ni_3_S_2_/NF is the smallest among the series samples and suggests its quick OER reaction kinetics. The charge transfer resistance (*R*_ct_) collected by electrochemical impedance spectroscopy (EIS) is also used to explore the electron-transfer kinetics of the catalyst. The EIS curves of a series of electrodes obtained under open circuit voltage are shown in Figure 6d. There is the largest solution resistance of pure Ni foam, and the solution resistance decreased effectively via grown Ni_3_S_2_ series samples on Ni foam. The semicircle diameter of Fe_2.3%_-Ni_3_S_2_/NF is the smallest among the five electrodes, and its calculated *R*_ct_ value is 1.57 Ω. As a comparison, the obtained *R*_ct_ values of Ni foam, Ni_3_S_2_/NF, Fe_0.6%_-Ni_3_S_2_/NF, Fe_0.8%_-Ni_3_S_2_/NF, and Fe_3.7%_-Ni_3_S_2_/NF are 19.73, 28.59, 2.67, 4.82, and 6.45 Ω, respectively. Ni_3_S_2_/NF obtained without Fe-Ni MOF precursor owns the largest *R*_ct_ value among the electrodes; nevertheless, the Fe-doped series of Ni_3_S_2_/NF electrodes synthesized by Fe-Ni MOF precursors display better conductivity. Furthermore, the results suggest that an appropriate amount of Fe ions doping can improve the charge transfer efficiency, which further demonstrates the highest OER activity of Fe_2.3%_-Ni_3_S_2_/NF among the series electrodes.

The electrochemical active surface area (ECSA) can reflect the real OER catalytic activity of the as-prepared samples. Within the non-Faraday region of voltage windows of 0~0.1 V (vs. Hg/HgO), the capacitance of the double electric layer C_dl_ value was calculated through CV testing with different scan rates. Appendix A shows the CV curves of the series electrodes at scanning rates ranging from 10 to 100 mV s^−1^, and C_dl_ was calculated by the ratio of the scan rate to the equation *j* = *j*_a_ − *j*_c_ at 0.05 V (vs. Hg/HgO). Figure 7a requires the slope of the line obtained after linear fitting of the current density at 0.974 V (vs. RHE) and the measured scan rate, which corresponds to the 2C_dl_ value of all the samples. The measured C_dl_ values of Ni_3_S_2_/NF, Fe_0.6%_-Ni_3_S_2_/NF, Fe_0.8%_-Ni_3_S_2_/NF, Fe_2.3%_-Ni_3_S_2_/NF, and Fe_3.7%_-Ni_3_S_2_/NF are 1.55, 2.20, 2.97, 3.57, and 2.81 mFcm^−2^, respectively (Figure 7a), which corresponds to the ECSA values of 38.75, 55, 74.25, 89.25, and 70.25 cm^2^. This result indicates that the ECSA value of Fe_2.3%_-Ni_3_S_2_/NF exhibits the largest electrochemical active surface area among all the five electrodes. The high ECSA value of Fe_2.3%_-Ni_3_S_2_/NF may be related to the incorporation of appropriate Fe ions and their unique morphology. After Fe ions doping, the morphology changes from tightly arranged nanosheets to vertically arranged dendritic structures. The unique morphology and proper Fe ions doping are conducive to exposing more active sites and effectively improving the OER activity.

Stability testing is also a key indicator for estimating the performance of OER electrocatalysts. Multicurrent steps chrono-potentiometric curves of all prepared electrodes were performed and are presented in Figure 7b. Figure 7b exhibited chrono-potentiometric responses at various current densities in the range of 10 mA cm^−2^ to 100 mA cm^−2^, and the potential remained unchanged in each increment when increased by 10 mA cm^−2^ every 500 s, demonstrating that the electrodes have high mass transportation efficiency. The potential of Fe_2.3%_-Ni_3_S_2_/NF in each step is lower than that of other electrodes, indicating that it has good stability under both low and high current density. Figure 7c shows the electrochemical stability test conducted at a constant voltage of 0.637 V (vs. Hg/HgO), and the current density curves of all electrodes showed no significant attenuation after 12 h of testing. LSV curves of the optimized Fe_2.3%_-Ni_3_S_2_/NF electrode before and after stability testing were compared and are shown in Figure 7d, and the polarization curves are almost coincident, further suggesting that Fe_2.3%_-Ni_3_S_2_/NF has excellent stability for OER. As seen from SEM images in Appendix A, the morphology of all Fe_x_-Ni_3_S_2_/NF series samples still remained dendritic after long-term stability.

Figure 8 shows the XPS plot of Fe_2.3%_-Ni_3_S_2_/NF before and after stability tests. It can be seen that the position of the Ni/Fe/S peaks has a low-amplitude positive shift (Appendix A). The Ni, Fe, and S signals were detected in the survey shown in Figure 8a, indicating the existence of Ni, Fe, and S elements in the Fe_2.3%_-Ni_3_S_2_/NF sample. The Ni 2p peaks in the high-resolution spectrum can be divided into two groups of doublets peaks centered at 852.5 (855.9 eV) and 869.7 eV (873.7 eV), indicating the coexistence of Ni (II) and Ni (III). After the stability test, the peak intensity of Ni (III) increased, indicating that Ni (III) may be the active site during the OER process. In the Fe 2p spectra, the peaks located at 706 and 713 eV corresponded to Fe (II) and Fe (III) orbitals, respectively. Furthermore, the satellite of S 2p located at ~170 eV increased, which indicated the partial oxidation of the S species on Fe_2.3%_-Ni_3_S_2_/NF during the long-term OER stability test process. In addition, the diffraction peaks in the PXRD pattern showed little change after OER stability testing (Appendix A). All the test results of SEM, XPS, and PXRD demonstrate the good stability of the Fe_2.3%_-Ni_3_S_2_/NF electrode and promise its potential practical application in the future.

## 4. Conclusions

In summary, a series of Ni_3_S_2_ dendritic array structures doped with different trace Fe elements on NF were successfully synthesized in this work by the two-step solvothermal method. The catalysts were grown in situ on NF without using any binder and can be directly used as self-supporting electrodes. The optimized Fe_2.3%_-Ni_3_S_2_/NF electrode displayed excellent OER catalytic performance in an alkaline solution. The overpotential of the Fe_2.3%_-Ni_3_S_2_/NF electrode was only 233 mV at a current density of 10 mA cm^−2^, and the acquired overpotential was also small (378 mV), even at a high current density of 100 mA cm^−2^, better than many other reported similar electrodes. The corresponding Tafel slope of the electrode was 66 mV dec^−1^ and revealed fast reaction kinetics. The current density almost did not decay during the stability test for 12 h. The introduction of trace Fe element into Ni_3_S_2_ promotes the exposure of active sites by regulating the morphology of the array electrocatalyst, and the synergistic effect between Fe and Ni ions improves the intrinsic activity by combining with the conductive substrate. This work may provide an innovative method for the synthesis of low-cost micro-/nano-structured transition metal sulfide electrocatalysts to enhance OER performances.

## Data Availability

The raw data supporting the conclusions of this article will be made available by the authors on request.

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
