# Peer review of "Nanoarchitectonics of Fe-Doped Ni3S2 Arrays on Ni Foam from MOF Precursors for Promoted Oxygen Evolution Reaction Activity"

_nanomaterials, 2024, doi:10.3390/nano14171445_

Round 1

Reviewer 1 Report

Comments and Suggestions for Authors

The authors prepared Fe doped Ni3S2  catalyst supported by Ni-foam (NF), by a hydrothermal procedure. The characterization by contemporary techniques confirmed the increased content of dopant, and an uniform deposition of electrocatalyst on the NF threads. The authors used these electrocatalysts to study  Oxygen evolution reaction (OER) in alkaline medium (1M KOH). The increase of dopant content to some extent enhanced the catalytic activity.  This is an interesting result for expert readers, however, some improvements are needed, mostly in order to provide comparability of the results of this study with the already published  data:

The objections:

1.       Section 2.2.  The second paragraph of this section contain almost identical description of FeNi-MOF/NF preparation appearing also in the first paragraph. That looks like confusing to the readers.

2.       Section 2.4.. and related content of the Results and Conclusions:  Some basic information about the working electrode are missing:  the thickness, the porosity, the mean thread diameter and mass per square cm of Ni-foam; what is the mass loading of catalyst?  How the electrode reactions on the working electrode surface were protected from the influence of contacting wire? Furthermore, a clear information is needed what is the meaning of the electrode surface area used to calculate current density in Fig’s 6 and 7, and in Table 1 – whether is that an apparent (geometric) surface area, or is a real surface area?  The comparisons in Table 1 assume  that the current densities are calculated on the basis of real surface areas, otherwise, the comparison loses its meaning.

3.        Fig 6(a). Please, mark the location of the zero overvoltage potential. Apart of that, it sounds reasonable to draw the OER polarogram for pure Ni-foam. As expected from already  published studies (see cited reference 22, p 39153) this curve should display lower activity than the Ni3S2/NF one?    

Author Response

Reviewer 1 Comments and Suggestions for Authors

The authors prepared Fe doped Ni3S2 catalyst supported by Ni-foam (NF), by a hydrothermal procedure. The characterization by contemporary techniques confirmed the increased content of dopant, and an uniform deposition of electrocatalyst on the NF threads. The authors used these electrocatalysts to study oxygen evolution reaction (OER) in alkaline medium (1M KOH). The increase of dopant content to some extent enhanced the catalytic activity.  This is an interesting result for expert readers, however, some improvements are needed, mostly in order to provide comparability of the results of this study with the already published data:

The objections:

  1. Section 2.2.  The second paragraph of this section contain almost identical description of FeNi-MOF/NF preparation appearing also in the first paragraph. That looks like confusing to the readers.

Response: In Section 2.2, the description of FeNi-MOF/NF preparation has been revised, and we have weakened the description of FeNi-MOF/NF preparation in the revised version.

  1. Section 2.4. and related content of the Results and Conclusions:  Some basic information about the working electrode are missing:  the thickness, the porosity, the mean thread diameter and mass per square cm of Ni-foam; what is the mass loading of catalyst?  How the electrode reactions on the working electrode surface were protected from the influence of contacting wire? Furthermore, a clear information is needed what is the meaning of the electrode surface area used to calculate current density in Fig’s 6 and 7, and in Table 1 – whether is that an apparent (geometric) surface area, or is a real surface area?  The comparisons in Table 1 assume that the current densities are calculated on the basis of real surface areas, otherwise, the comparison loses its meaning.

Response: The information of the working electrode was provided in the revised manuscript. The thickness of the Ni-foam is 1.6 mm, the average pore size is 0.2 mm, cut to 1*3 cm2, and mass per square cm of Ni-foam is ~34 mg. The mass loading of catalyst is ~1.2 mg cm-2, the electrode was fixed in Pt electrode clamp and the electrode reactions on the working electrode surface were protected from the influence of contacting wire. An apparent (geometric) surface area used to calculate current density in Figures 6 and 7, and in Table 1. The comparisons in Table 1 listed the current densities are calculated based on geometric surface areas.

  1. Fig 6 (a). Please, mark the location of the zero overvoltage potential. Apart of that, it sounds reasonable to draw the OER polarogram for pure Ni-foam. As expected from already published studies (see cited reference 22, p 39153) this curve should display lower activity than the Ni3S2/NF one?   

Response: The OER polarogram for pure Ni-foam has been provided in the revised manuscript, and the OER activity of Ni-foam is lower than the Ni3S2/NF.

Reviewer 2 Report

Comments and Suggestions for Authors

Although research targets may have some routine (less innovative) aspects, this work really well collects necessary data Research level itself is high. Publication of these data in public journal media would have good contributions to the related research fields. From these positive viewpoints, I may suggest publication of this work in Nanomaterials with necessary revisions. I suggest several revisions. Please see below.

1) Discussions on Table 1, Compared with ....... Ni3S2/NF series electrodes, are fine. However, more quantitative descriptions using concrete values would make this part much stronger.

2) Inclusion of new concept words in the title often effectively increase innovative impression to the work. I may suggest use of an emerging conceptual term, nanoarchitectonics, in the title (for the concept, see https://academic.oup.com/bcsj/article/97/1/uoad001/7457599). For example, the title like ... Nanoarchitectonics of Fe-doped Ni3S2 Arrays on Ni Foam from MOF Pre-2 cursors for Promoted Oxygen Evolution Reaction Activity ... may sound more innovative.

3) Although this work is well done, conclusion description is rather insufficient. In addition to summary descriptions, more detailed descriptions on future perspectives with possible applications can be added to increase innovative impression.

4) Scheme 1 is too schematics (not highly scientific). More information based on chemical structures has to be added.

5) References are generally good but can be more updated and generalized adding comprehensive reviews to the initial part of Introduction (for example, see https://academic.oup.com/bcsj/article/95/1/73/7226604, https://pubs.rsc.org/en/content/articlelanding/2024/cs/d3cs00723).

6) Words (length notifications) on scale bars are not well visible because of white color overlapping. Please use different colors.

Author Response

Reviewer 2 Comments and Suggestions for Authors

Although research targets may have some routine (less innovative) aspects, this work really well collects necessary data Research level itself is high. Publication of these data in public journal media would have good contributions to the related research fields. From these positive viewpoints, I may suggest publication of this work in Nanomaterials with necessary revisions. I suggest several revisions. Please see below.

1) Discussions on Table 1, Compared with ....... Ni3S2/NF series electrodes, are fine. However, more quantitative descriptions using concrete values would make this part much stronger.

Response: Discussions on Table 1, more quantitative descriptions using concrete values was compared in the revised version.

2) Inclusion of new concept words in the title often effectively increase innovative impression to the work. I may suggest use of an emerging conceptual term, nanoarchitectonics, in the title (for the concept, see https://academic.oup.com /bcsj/article/97/1/uoad001/7457599). For example, the title like ... Nanoarchitectonics of Fe-doped Ni3S2 Arrays on Ni Foam from MOF Precursors for Promoted Oxygen Evolution Reaction Activity ... may sound more innovative.

Response: The title was revised as the reviewer’s suggestion, “Nanoarchitectonics of Fe-doped Ni3S2 Arrays on Ni Foam from MOF Precursors for Promoted Oxygen Evolution Reaction Activity” (Ariga, K. Nanoarchitectonics: the method for everything in materials science, B. Chem. Soc. Jpn., 2024, 97, uoad001).

3) Although this work is well done, conclusion description is rather insufficient. In addition to summary descriptions, more detailed descriptions on future perspectives with possible applications can be added to increase innovative impression.

Response: The conclusion description was revised, and some detailed descriptions of possible applications has been added in the revised manuscript.

4) Scheme 1 is too schematics (not highly scientific). More information based on chemical structures has to be added.

Response: Scheme 1 was redrawn in the revised version, and more information based on chemical structures has been added.

5) References are generally good but can be more updated and generalized adding comprehensive reviews to the initial part of Introduction (for example, see https://academic.oup.com/bcsj/article/95/1/73/7226604, https://pubs.rsc.org/en/content/articlelanding/2024/cs/d3cs00723).

Response: The above comprehensive review article has been cited in the revised Introduction section. (B. Chem. Soc. Jpn., 2022, 95, 73–103)

6) Words (length notifications) on scale bars are not well visible because of white color overlapping. Please use different colors.

Response:  Words on scale bars in Fig.2 and Fig.3 were all marked with a black stroke and white fill style in the revised version.

Reviewer 3 Report

Comments and Suggestions for Authors

The authors have prepared various Fe-doped Ni3S2 catalysts by solvohermal synthesis process and applied the catalysts to OER reaction. The research work presented is comprehensive and the results are also interesting. The manuscript can accepted after clarifying some of the point as below.

In the introduction the authors should describe the technical drawbacks of Ni3S2 as such and why secondary metal doping is considered an effective strategy, particularly why Fe. What advantages that it brings to? Authors should also bring to the kind notice of the reader about the Ni3S2 morphological influence on the electrocatalytic activity in the introduction section. Overall, the introduction should be more clear why authors choose Ni3S2.

Experimental Section, especially the Electrochemical Measurements section, should describe how the electrochemical studies have been conducted, starting from the ink preparation and how the material has been coated, loading of the catalysts, potential range, impedance potential, frequency range, stability test related and lot more. Authors should know the experimental procedure in order to repeat the experiments of their interest.

XRD peaks must be indexed with Ni3S2 phases and their planes, both in the text and in the figure 1.

Authors should be able to describe the morphological changes to Ni3S2 with the introduction of Fe and sulfurization in SEM image.

In the manuscript line 153 – “only Fe ions were added during the first solvothermal step, and the  second step was sulfurization treatment normally”. The authors mentioned here that the Fe is added in the first step and S in the second. However, the schematic shows, Fe is added in all the two steps. Why?oqH

Why and what is the reason for low Fe content in the first solvothermal process>?

In the figure 6 a, could you please explain why is the redox peak shifting to the higher potentials? And its significance. Also 6c, authors should calculate the tafel slope for all the catalysts at the same current density values. In the tafel slope, need not to give values after the decimal point.

Are there any shifts in the peaks of the Ni/Fe/S after the stability test? Its strongly suggested to tabulate the XPS peak values before and after the stability test.

Comments on the Quality of English Language

Moderate English corrections are required.

Author Response

Reviewer 3 Comments and Suggestions for Authors

The authors have prepared various Fe-doped Ni3S2 catalysts by solvothermal synthesis process and applied the catalysts to OER reaction. The research work presented is comprehensive and the results are also interesting. The manuscript can accepted after clarifying some of the point as below.

1) In the introduction the authors should describe the technical drawbacks of Ni3S2 as such and why secondary metal doping is considered an effective strategy, particularly why Fe. What advantages that it brings to? Authors should also bring to the kind notice of the reader about the Ni3S2 morphological influence on the electrocatalytic activity in the introduction section. Overall, the introduction should be more clear why authors choose Ni3S2.

Response: The reason of chosen Ni3S2 structure was added in the revised version. Ni3S2 with a hexagonal structure constructed by Ni-Ni metal bonds have been reported exhibit the similar electrocatalytic OER performance to RuO2, however, the overpotential of pure Ni3S2 is always high because of its sluggish OER kinetics. Transition metal ions doping is an effective way to enhance the intrinsic activity, especially, the Fe-Ni bimetallic synergistic effect contributed to formation key active intermediates (FeOOH-NiOOH). Through Fe ions doping not only can adjust the surface electron distribution, but also can increase the charge transfer kinetics. In this work, Fe-doped Ni3S2 branch arrays have been designed as high effective OER catalysts mainly due to their branch structure, which can improve the electrochemical surface area, conductivity and enhance the long-time catalytic stability (J. Mater. Chem. A, 2023, 11, 5734–5745;

Inorg. Chem. Front., 2022, 9, 6237–6247).

2) Experimental Section, especially the Electrochemical Measurements section, should describe how the electrochemical studies have been conducted, starting from the ink preparation and how the material has been coated, loading of the catalysts, potential range, impedance potential, frequency range, stability test related and lot more. Authors should know the experimental procedure in order to repeat the experiments of their interest.

Response:  In the electrochemical measurements section, the self-standing Fe-doped Ni3S2/NF electrodes were used as working electrodes directly, avoiding from the ink preparation and drop-coating process. The loading weight of the catalysts is ~1.2 mg cm-2, in the electrochemical measurements section, the potential range is 0-0.8 V vs. Hg/HgO, the impedance potential is kept on open circuit potential and the frequency range is 0.1 Hz-1 MHz. The stability test carried out at a potential of 0.637 V vs. Hg/HgO for 12 h.

3) XRD peaks must be indexed with Ni3S2 phases and their planes, both in the text and in the figure 1.

Response:  The XRD peaks have been indexed with Ni3S2 phases and their planes both in the text and in the Figure 1.

4) Authors should be able to describe the morphological changes to Ni3S2 with the introduction of Fe and sulfurization in SEM image.

Response: The morphological changes to Ni3S2 with the introduction of Fe and sulfurization in SEM image were described in the revised version. Xu et al. reported Fe3+ substituted Ni ions in the (101) plane of Ni3S2 and promote the (101) plane grown in [001] direction (Sustain. Energy Fuels, 2020, 4, 3326−3333). Thus, the introduce Fe ions in Ni3S2 lattice will promote the formation of branch morphology, which have been supported by the result of HRTEM results with the exposing of (101) crystallographic planes.

5) In the manuscript line 153 – “only Fe ions were added during the first solvothermal step, and the second step was sulfurization treatment normally”. The authors mentioned here that the Fe is added in the first step and S in the second. However, the schematic shows, Fe is added in all the two steps. Why?

Response:  For the synthesis of Fe0.6%-Ni3S2/NF, only Fe ions were added during the first solvothermal step, and the second step was sulfurization treatment normally. However, to obtain the desired electrode with a higher Fe ions content from 0.8% to 3.7%, Fe ions must be added in all the two steps, which was illustrated in Scheme 1.

6) Why and what is the reason for low Fe content in the first solvothermal process?

Response: Compared with Ni ions, Fe ions and carboxylic acid ligand of 1,2,4,5-benzenetetra-carboxylic acid are difficult to form MOF precursors.

7) In the figure 6 a, could you please explain why is the redox peak shifting to the higher potentials? And its significance. Also 6c, authors should calculate the tafel slope for all the catalysts at the same current density values. In the tafel slope, need not to give values after the decimal point.

Response:  In the Figure 6 a, the redox peak shifting to the higher potentials indicated the Fe incorporation in Ni3S2, the Fe ions incorporation increases the Ni redox potential and decreases the area of the Ni redox peaks (Nanoscale, 2019, 11, 8170–8184). The Tafel slope have been revised.

8) Are there any shifts in the peaks of the Ni/Fe/S after the stability test? Its strongly suggested to tabulate the XPS peak values before and after the stability test.

Response: All the peaks of the Ni/Fe/S after the stability test shift positively, and the XPS peak values the Ni/Fe/S before and after the stability test tabulated in supporting information as Table S1.

Round 2

Reviewer 1 Report

Comments and Suggestions for Authors

In the response to my objection No 2, the authors stated:  An apparent (geometric) surface area used to calculate current density in Figures 6 and 7, and in Table 1.

However, this sentence was not introduced in the text of manuscript. This data is important, and should be located  in Section 2.4, or in line 197/198

Author Response

Reviewer 1 Comments and Suggestions for Authors

In the response to my objection No 2, the authors stated:  An apparent (geometric) surface area used to calculate current density in Figures 6 and 7, and in Table 1.

However, this sentence was not introduced in the text of manuscript. This data is important, and should be located in Section 2.4, or in line 197/198.

Response: We added the related information of geometric surface area in Section 2.4 in the revised manuscript.

Reviewer 3 Report

Comments and Suggestions for Authors

After reviewing the responses to my comments, I found that the authors have addressed them thoroughly and satisfactorily. Therefore, I believe the manuscript is suitable for publication in its current form.

Comments on the Quality of English Language

Fine

Author Response

Reviewer 3 Comments and Suggestions for Authors

After reviewing the responses to my comments, I found that the authors have addressed them thoroughly and satisfactorily. Therefore, I believe the manuscript is suitable for publication in its current form.

Response:  We have revised the manuscript again, and some minor mistakes were corrected.